# Learning Fine-Grained Representations through Textual Token Disentanglement in Composed Video Retrieval

**Yue Wu**[1,2,4], **Zhaobo Qi**[3], **Yiling Wu**[2], **Junshu Sun**[1], **Yaowei Wang**[2,5], **Shuhui Wang**[1,2*]
[1] Key Lab of Intell. Info. Process., Inst. of Comput. Tech., CAS
[2] Pengcheng Laboratory   [3] Harbin Institute of Technology, Weihai
[4] University of Chinese Academy of Sciences   [5] Harbin Institute of Technology, Shenzhen
wuyue221@mails.ucas.ac.cn   qizb@hit.edu.cn
{sunjunshu21s, wangshuhui}@ict.ac.cn   wangyw@pcl.ac.cn

## Abstract

With the explosive growth of video data, finding videos that meet detailed requirements in large datasets has become a challenge. To address this, the composed video retrieval task has been introduced, enabling users to retrieve videos using complex queries that involve both visual and textual information. However, the inherent heterogeneity between the modalities poses significant challenges. Textual data are highly abstract, while video content contains substantial redundancy. The modality gap in information representation makes existing methods struggle with the modality fusion and alignment required for fine-grained composed retrieval. To overcome these challenges, we first introduce **FineCVR-1M**, a fine-grained composed video retrieval dataset containing 1,010,071 video-text triplets with detailed textual descriptions. This dataset is constructed through an automated process that identifies key concept changes between video pairs to generate textual descriptions for both static and action concepts. For fine-grained retrieval methods, the key challenge lies in understanding the detailed requirements. Text description serves as clear expressions of intent, but it requires models to distinguish subtle differences in the description of video semantics. Therefore, we propose a textual Feature Disentanglement and Cross-modal Alignment framework (**FDCA**) that disentangles features at both the sentence and token levels. At the sequence level, we separate text features into retained and injected features. At the token level, an Auxiliary Token Disentangling mechanism is proposed to disentangle texts into retained, injected, and excluded tokens. The disentanglement at both levels extracts fine-grained features, which are aligned and fused with the reference video to extract global representations for video retrieval. Experiments on FineCVR-1M dataset demonstrate the superior performance of FDCA. Our code and dataset are available at: https://may2333.github.io/FineCVR/.

## 1 Introduction

The explosive growth of video data poses a challenge for users seeking videos that meet fine-grained requirements within a vast video database. This demand can be modeled as complex query patterns, *i.e.*, composed video retrieval (CVR) (Ventura et al., 2024; Hummel et al., 2024). CVR combines a reference video and a modification text as a query to find visually similar and semantically relevant videos with altered objects, attributes, or actions. However, existing methods (Ventura et al., 2024; Hummel et al., 2024), including foundational visual-language models(Radford et al., 2021; Li et al., 2022), perform poorly in meeting fine-grained retrieval requirements (Ventura et al., 2024). We identify the primary reason is the semantic granularity gap between textual and visual representations. To address this, we first introduce a new dataset with more detailed textual descriptions. Second, we decompose the textual representation for fine-grained fusion and alignment.

---

*Corresponding author.

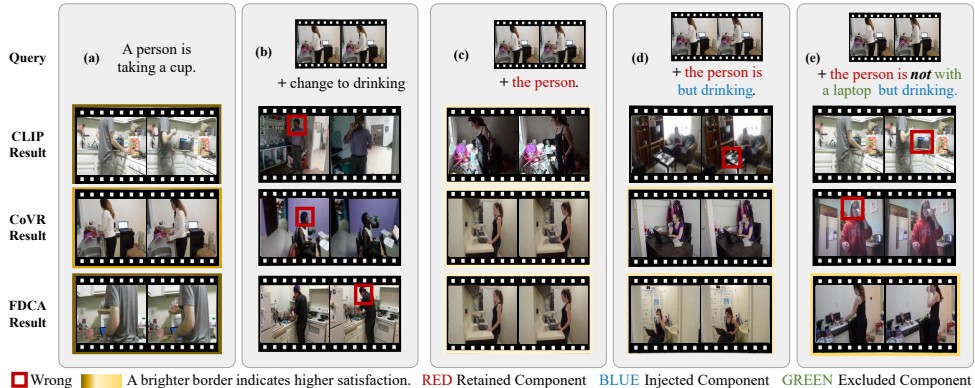

Figure 1: A user wants to retrieve videos of similar people "taking a cup not with a laptop and drinking". (a) In video-text retrieval, the results often fail to meet user expectations from a visual perspective due to the limitations of text queries. (b) In composed video retrieval, focusing on the altered elements is vague, leading to mismatched results. (c)-(e) As the three components (in different colors) are gradually incorporated into the modification text, the retrieved videos become increasingly aligned with the user's intent.

From the dataset perspective, the current CVR datasets include CoVR (Ventura et al., 2024) and EgoCRV (Hummel et al., 2024). However, these datasets fail to convey the fine-grained user demands, whereas the modification texts focus solely on the changed element, neglecting to specify the visual information that should remain consistent. This oversight often causes redundant visual information in the reference videos to mislead models. Consequently, the returned videos may contain incorrect visual parts. See the example in Figure 1 (b), the phrase "change to drinking" highlights the action of drinking but fails to convey the essential visual context from the reference video. As a result, the returned videos either visually resemble the reference video or simply depict drinking, failing to deliver satisfactory results. Therefore, it is essential to introduce more precise modification texts that explicitly convey fine-grained requirements to enhance retrieval effectiveness.

To facilitate the fine-grained CVR task while minimizing extensive annotating efforts, we highlight the conceptual similarities and differences in text queries and construct a new dataset **FineCVR-1M**, containing over one million triplets. Each triplet consists of a pair of similar videos and modification text that precisely describes the conceptual relevance between the videos. To convey the fine-grained user demands, we generate modification texts based on three components, including **retained component** that remain consistent in the reference and target videos, **injected component** that are newly added in the target videos, and **excluded component** that are expected to be excluded from the target videos. Compared to the simple phrases (Liu et al., 2021; Wu et al., 2021; Ventura et al., 2024; Hummel et al., 2024), modification texts with the three components explicitly express fine-grained user demands on complex video content. FineCVR-1M constitutes a total of 1,010,071 triplets encompassing diverse concepts. It can be utilized for developing CVR models and enhancing the fine-grained information-seeking ability of multimodal foundation models.

Given existing coarse-grained CVR datasets, current CVR methods (Ventura et al., 2024; Hummel et al., 2024; Thawakar et al., 2024) primarily focus on cross-modal alignment and fusion at the sentence level. This treatment not only leads to highly-coupled feature extraction and inferior performance on the fine-grained CVR task, but also lacks exploration on the finer-grained demands that can be well described by the above three language components. To facilitate fine-grained retrieval, we propose a Textual Feature Disentanglement and Cross-modal Alignment framework (FDCA). FDCA disentangles modification text at both sentence and token levels, where cross-modal alignment is performed between video and text as a basic objective function. By leveraging both sentence- and token-level semantic analysis, FDCA ensures a more accurate alignment between the text and visual content, ultimately improving the performance of fine-grained CVR.

We compare our method with several composed retrieval methods on the FineCVR-1M dataset. Experimental results demonstrate that our method outperforms existing methods by a clear margin. Our study provides strong support from both dataset and methodology perspectives in enhancing the

fine-grained information seek ability of multi-modal video pre-trained models. To summarize, our contribution is threefold:

- We construct a benchmark **FineCVR-1M** with 1M+ triplets. This benchmark supports the combined query with both reference videos and modification text for fine-grained video retrieval and is publicly available for download.

- We propose **FDCA** that performs text feature disentangling at sentence and token levels to progressively enhance the descriptive power of features of the reference video, facilitating efficient retrieval of target videos that visually and semantically satisfy user expectations.

- Extensive experiments show the high quality of the FineCVR-1M dataset and the advantage of FDCA.

## 2  RELATED WORK

**Composed Query Dataset** requires generating captions for the differences between the reference and target videos. Two CVR datasets have recently emerged, *i.e.*, CoVR (Ventura et al., 2024) and EgoCRV (Hummel et al., 2024). CoVR employs a large language model (LLM) (Touvron et al., 2023a) to generate captions highlighting the differences between two reference captions, while EgoCRV manually creates a test set for first-person perspective videos. However, the modification texts in both datasets do not explicitly describe fine-grained requirements, leading to an inability in supporting fine-grained CVR tasks. In our work, we construct datasets by employing clear textual prompts and the LLM to automatically generate detailed text descriptions based on video content, thereby well supporting fine-grained composed retrieval.

**Composed Image Retrieval** (CIR) aims to get the target image by a composed query including a reference image and modification text. Current CIR models focus mainly on feature discrimination and feature composition. Discriminating features requires distinguishing the retained features in the reference images, and the features need to be injected into the target image. Existing methods employ gate mechanism (Vo et al., 2019) or semantic-guided attention mechanism (Chen et al., 2020b; Lee et al., 2021; Wen et al., 2021; Hosseinzadeh & Wang, 2020; Baldrati et al., 2023; Saito et al., 2023) to discriminate these features. The feature composition is addressed by projecting visual and textual features into a common space, typically utilizing a cross-modal Transformer (Saito et al., 2023; Baldrati et al., 2023). However, due to the lack of temporal information, these CIR methods can only be applied to static modifications and cannot be extended to CVR directly.

**Composed Video Retrieval** (CVR) extends the task of composed retrieval from the image domain to the video domain. The CoVR (Ventura et al., 2024) employ the BLIP (Li et al., 2022) directly to achieve cross-modal alignment and fusion, while EgoCVR uses a training-free reranking approach. To obtain the enhanced semantics from modification text, (Thawakar et al., 2024) leverages an LLM to enrich the modification texts. However, these methods emphasize coarse-level feature alignment. In contrast, our FDCA focuses on discriminating textual features into three meaningful components for fine-grained alignment.

## 3  FINECVR-1M DATASET CONSTRUCTION

Items in the dataset are organized as triplets $\{\mathcal{V}_r, \mathcal{T}, \mathcal{V}_t\}$, where $\mathcal{T}$ denotes the modification text describing the differences between a reference video $\mathcal{V}_r$ and a target video $\mathcal{V}_t$. We first match similar videos as the reference video and the target video, then generate modification texts automatically. For the static concepts, modification texts are generated by filling different and similar key static concepts into the textual prompts. For action concepts, we directly employ a fine-tuned LLM to generate descriptions of the action differences and similarities.

### 3.1  VIDEO PREPROCESSING AND PAIRING

To ensure diversity and accuracy, we first collect videos from easily accessible datasets. We use four datasets, ActionGenome (Ji et al., 2020), ActivityNet (Fabian Caba Heilbron & Niebles, 2015), HVU (Diba et al., 2020), and MSRVTT (Xu et al., 2016) as our video source. These videos provide rich information on the visual contents, such as actions, scenes, objects, and attributes. To reduce the redundancy of the videos, we clip videos into different events based on their temporal boundary

Table 1: Statistics of FineCVR-1M and existing CVR datasets. "RC" denotes "Retained Component", "IC" denotes "Injected Component", and "EC" denotes "Excluded Component". FineCVR-1M incorporates more diversified concept types and fine-grained components in the descriptions.

| | Train triplets | Test triplets | Visuals | Unique word | Attribute-Centric | Object-Centric | Scene-Centric | Action-Centric | RC | IC | EC |
|---|---|---|---|---|---|---|---|---|---|---|---|
| WebVid-CoVR | 1,648,789 | 2,556 | 133,219 | 21,098 | - | 85.0% | - | 15.0% | | ✓ | |
| EgoCVR | - | 2,295 | 2,295 | 940 | - | 21.1% | - | 78.9% | | ✓ | |
| FineCVR-1M | 1,000,028 | 10,043 | 136,547 | 20,961 | 0.5% | 22.1% | 28.5% | 48.9% | ✓ | ✓ | ✓ |

annotations and remove videos lacking corresponding captions. To match similar video pairs that contain slightly different content, following (Chen et al., 2020a; Xu & Wang, 2021), we compute the cosine similarity among the BLIP-2 features (Li et al., 2023) of videos and select the top 20 matches for each video. Pairing videos facilitates the comparison of video content and generating their corresponding modification texts.

## 3.2 MODIFICATION TEXT GENERATION

### 3.2.1 STATIC CONCEPTS

**Basic Static Concept Detection.** Static concepts, in contrast with the action concepts of the objects, refer to contents such as objects, attributes, and scenes. Basic static concepts in the videos serve as the flags for identifying both the content differences between videos and the contents of query interests. To ensure a comprehensive detection of the static concepts, we extract them from both video frames and their annotations. The former are often visually salient concepts, while the latter are of high-quality and semantically significant.

**Key Static Concept Selection.** In this step, we assign scores for concepts of each video clip to identify the key concepts. Specifically, concepts from human annotations are considered highly credible and align with human attention, so they obtain higher scores. In contrast, concepts from video frames may contain error and noise, so they obtain lower scores. Based on the assigned scores, we select the key static concepts with the top 5 highest scores in each type for every video. An example is shown in the Appendix Figure A9.

**Modification Text Generation.** We compare the key static concepts between video pairs and generate descriptions of their differences with three types of prompts. Details are in the Appendix B.2.

### 3.2.2 ACTION CONCEPTS

In addition to the static video concepts, we also generate modification texts regarding the fine-grained action differences by fine-tuning LLaMA2 (Touvron et al., 2023b). Specifically, we construct a prompt (further elaborated in Appendix B.2) to instruct the ChatGPT (Ouyang et al., 2022) to generate text that describes the action differences between caption annotations. The generated descriptions are manually corrected for the fine-tuning of LLaMA2 with LoRA (Hu et al., 2021). We apply the fine-tuned LLaMA2 to the matched similar video pairs and produce descriptions of the action differences between videos.

## 3.3 DATASET PROPERTY

FineCVR-1M consists of 1,000,028 triplets for training and 10,043 triplets for testing. Detailed statistics are presented in Table 1. Given the unavoidable noise in the automatic construction process, we manually select triples with accurate ground-truth modification text descriptions as the test subset for evaluation. Although the number of triplets is smaller than that of WebVid-CoVR, FineCVR-1M has a larger number of videos and comparable richness in the descriptive texts. In comparison with the other two CVR datasets that focus only on objects and actions, FineCVR-1M targets four categories (attribute, object, scene, and action) for modification and retains three crucial components (**retained component**, **injected component**, and **excluded component**) in the modification texts, enabling fine-grained video retrieval tasks. More analysis of FineCVR-1M are shown in Appendix B.3.

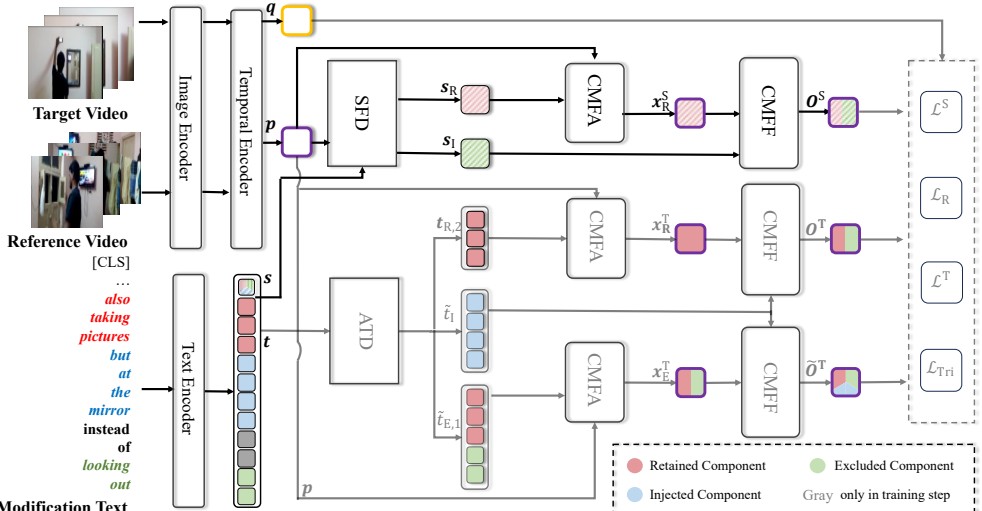

Figure 2: The pipeline of FDCA involves fine-grained cross-modal alignment and fusion through the disentangling of text features. We further enhance this process by introducing token-level disentangling, where clustering is used to generate three types of features, enabling the model to focus on fine-grained information.

## 4 METHOD

Given a pair of reference video and modification text $\{\mathcal{V}_r, \mathcal{T}\}$ as query, CVR requires cross-modal alignment to relate the composed query fusion feature with a video feature. This fusion feature can be employed to retrieve the target video $\mathcal{V}_t$ from the video database, which should describe both visual and semantic information. As shown in Figure 2, to capture the fine-grained user demand, we propose a Textual Feature Disentanglement and Cross-modal Alignment framework (FDCA), disentangling modification text at both sentence and token levels. Specifically, Sentence-level Feature Disentangling (SFD) aims to disentangle the text feature into the retained and injected components, followed by Cross-modal Feature Alignment (CMFA) and Cross-modal Feature Fusion (CMFF). At the token level, we introduce Auxiliary Token Disentangling (ATD) during training. ATD clusters the token-level features, and disentangles tokens into three types of components. These disentangled tokens are used in three auxiliary loss functions to guide the alignment and fusion process, resulting in a semantically enriched video feature for more accurate retrieval.

### 4.1 SENTENCE-LEVEL DISENTANGLEMENT AND FUSION

The SDF aims to obtain a global fused video feature through modification text feature disentangling and cross-modal fusion.

**Video and Text Encoding.** Given a video with $f$ frames, we first employ an image encoder (Radford et al., 2021; Li et al., 2022) to extract the reference frame features. The features are then fed into a temporal encoder and integrated as the reference video feature $\boldsymbol{p} \in \mathbb{R}^{1 \times d}$. For the modification text of length $l$ that corresponds to the video, a text encoder (Radford et al., 2021; Li et al., 2022) is deployed to extract the sentence-level text feature $\boldsymbol{s} \in \mathbb{R}^{1 \times d}$.

**Sentence-level Feature Disentangling.** Based on the semantic similarity between the modification text and reference video, we design a cross-attention strategy that is guided by the reference video feature. It extracts the retained component from the modification text regarding the reference video. Specifically, we project $\boldsymbol{p}$ and $\boldsymbol{s}$ into the latent space as query, key, and value: $\boldsymbol{Q} = \Psi_Q(\boldsymbol{p}), \boldsymbol{K} = \Psi_K(\boldsymbol{s}), \boldsymbol{V} = \Psi_V(\boldsymbol{s})$, where $\Psi_Q, \Psi_K, \Psi_V$ are implemented as linear layers. Then, the retained text feature $\boldsymbol{s}_R \in \mathbb{R}^{1 \times d}$ is calculated by the reference-guided cross-attention $\boldsymbol{s}_R = \text{softmax}(\frac{\boldsymbol{Q}\boldsymbol{K}^\top}{\sqrt{d}})\boldsymbol{V} + \boldsymbol{V}$. We assign the remaining features in $\boldsymbol{s}$ as injected text feature $\boldsymbol{s}_I = \boldsymbol{s} - \boldsymbol{s}_R$, which is the newly added semantic content in the target video.

**Cross-modal Feature Fusion.** The target video feature should capture the retrained feature in the reference video and fuse with the injected text feature. To align the retained text feature with the reference video feature, we input the concatenation of $p$ and $s_R$ into the Cross-modal Feature Alignment module, implemented by a Transformer encoder. This module captures the sentence-level retained component feature $x_R^S$ for the reference video. To further fuse with the injected text feature, the Cross-modal Feature Fusion, achieved by another Transformer encoder and a linear layer, integrates $x_R^S$ and $s_I$ to produce the global fused video feature $o^S$.

**Metric Learning.** We employ the contrastive loss to measure the distance between the global fused video feature $o^S$ and candidate video features in the database:

$$\mathcal{L}^S = -\frac{1}{|\mathcal{B}|} \sum_{i \in \mathcal{B}} \log \frac{\exp\left(o_i^S q_i^\top\right)}{\sum_{j \in \mathcal{B}} \exp\left(o_i^S q_j^\top\right)}, \tag{1}$$

where the $q_i$ is the candidate video feature. $\mathcal{B}$ is the batch set.

### 4.2 TOKEN-LEVEL DISENTANGLEMENT AND FUSION

Compared to the highly-coupled global sentence-level features, token features with more specific concept information are easier to disentangle. Shown in Figure 3, we use clustering to separate token features into retained, injected, and excluded components, enabling more effective capture of the fine-grained details.

**Token Feature Extraction.** The token feature $t \in \mathbb{R}^{l \times d}$ is encoded with the text encoder from the original text. Moreover, we also generate positive text by removing negation words like "instead of". Given the positive texts, the text encoder gives rise to the positive text feature $\tilde{t} \in \mathbb{R}^{(l-n) \times d}$, where $n$ denotes the number of negation words.

**Auxiliary Token Disentangling.** To distinguish the encoded token-level features as different components, we adopt the Density Peaks Clustering based on K-Nearest-Neighbors (DPC-KNN) (Du et al., 2016), a robust clustering technique commonly utilized for tokens clustering (Jin et al., 2023a;b; Zeng et al., 2022). Specifically, given tokens of the modification text, to ensure tokens in each cluster have the same component type, we first apply a one-dimensional convolutional layer across the word sequence. Then we use DPC-KNN to identify the two centers of tokens. The remaining tokens are classified into two sets based on the Euclidean distance from cluster centers. We then average the tokens within each cluster to derive two cluster features.

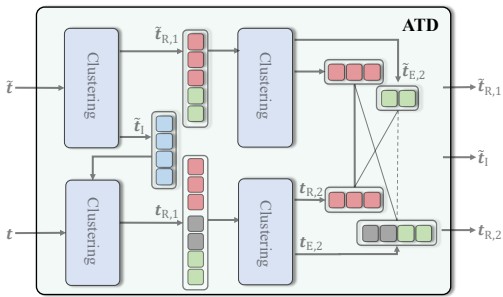

Figure 3: The ATD mechanism.

To identify the injected component, we first note that compared to the original modification text, the positive text is regarded to carry affirmative descriptions, and is thus more semantically similar to the reference video. Therefore, during the component disentanglement at the token level, we apply clustering to the positive tokens $\tilde{t}$ under the guidance of the reference video feature $p$. The clustering gives rise to the positive cluster features $\tilde{f}_1 \in \mathbb{R}^{2 \times d}$ of the two centers. The features of the cluster centers are then aggregated based on their similarity with the reference video feature $p$. The aggregation operation guided by the reference video learns the initial retained token feature $\tilde{t}_{R,1}$ from the positive text:

$$\tilde{t}_{R,1} = \text{softmax}(\beta \text{sim}(p, \tilde{f}_1)) \cdot \tilde{f}_1, \tag{2}$$

where $\text{sim}(\cdot)$ is dot product similarity, $\text{softmax}$ with $\beta$ is used to compute a soft index of the maximum similarity. Then we can obtain the injected token feature from the positive text as $\tilde{t}_I = \tilde{t} - \tilde{t}_{R,1}$.

For obtaining the retained and excluded components, note that initial retained tokens generated from the reference video may contain excluded tokens. To address this, we further disentangle the initial retained tokens and get the retained and excluded token features. Specifically, the positive texts and the original texts contain reciprocal information about the excluded component. We can first obtain the initial retained tokens in the original text and cluster these tokens into two clusters. By comparing

the clusters between the original text and the positive text, the most similar pair of clusters belongs to the retained component while the other pair belongs to the excluded component.

Following the clustering on the positive text, we also obtain the cluster features $\boldsymbol{f}_1 \in \mathbb{R}^{2 \times d}$ on the original text. The original modification text has similar injected components with its positive counterpart. Therefore, the injected token features obtained from the positive texts $\tilde{\boldsymbol{t}}_I$ can replace the injected features of the original text. Then, we can extract the initial retained token features $\boldsymbol{t}_{R,1}$ from $\boldsymbol{f}_1$ through:

$$\boldsymbol{t}_{R,1} = \left[1 - \text{softmax}\left(\beta \text{sim}\left(\tilde{\boldsymbol{t}}_I, \boldsymbol{f}_1\right)\right)\right] \cdot \boldsymbol{f}_1. \tag{3}$$

Next, we further cluster the initial retained tokens $\tilde{\boldsymbol{t}}_{R,1}$ and $\boldsymbol{t}_{R,1}$, resulting two pairs of clusters $\boldsymbol{f}_2 \in \mathbb{R}^{2 \times d}$ and $\tilde{\boldsymbol{f}}_2 \in \mathbb{R}^{2 \times d}$. Each constitutes a cluster of clean retained tokens and a cluster of excluded tokens. The clusters of clean retained tokens are more similar to each other. In contrast, the cluster of excluded tokens from the original text is accompanied by negation words, reciprocal to that from the positive text. To distinguish the clusters, similarities between the pairs of clusters are compared, resulting in the clean retained tokens $\boldsymbol{t}_{R,2}$ and the excluded tokens $\tilde{\boldsymbol{t}}_{E,2}$:

$$\boldsymbol{t}_{R,2} = \sum_{i=1}^{2} \text{softmax}(\beta \text{sim}(\boldsymbol{f}_2, \tilde{\boldsymbol{f}}_2))_i \cdot \boldsymbol{f}_2, \tag{4}$$

$$\tilde{\boldsymbol{t}}_{E,2} = \left[1 - \sum_{i=1}^{2} \text{softmax}(\beta \text{sim}(\boldsymbol{f}_2, \tilde{\boldsymbol{f}}_2))_i\right] \cdot \tilde{\boldsymbol{f}}_2, \tag{5}$$

where $i$ is the index of the two clusters.

**Auxiliary Loss Construction.** Given the disentangled fine-grained components at the token level, we further propose an auxiliary loss to ensure that the model focuses on fine-grained information. Resembling the sentence-level feature alignment and fusion, we also apply the CMFA and CMFF modules to the token-level features. For the feature alignment, CMFA aligns clean retained tokens $\boldsymbol{t}_{R,2}$ with the reference video, giving rise to the token-level retained component feature $\boldsymbol{x}_R^T$. For the feature fusion, CMFF fuses $\boldsymbol{x}_R^T$ with the injected positive tokens $\tilde{\boldsymbol{t}}_I$, generating the fused video feature $\boldsymbol{o}^T$. Similar to the Equation 1, the token-level contrastive loss $\mathcal{L}^T$ is calculated between the token-level fused feature $\boldsymbol{o}^T$ with the candidate video feature $\boldsymbol{q}$ to learn fine-grained token-level features as follows:

$$\mathcal{L}^T = -\frac{1}{|\mathcal{B}|} \sum_{i \in \mathcal{B}} \log \frac{\exp\left(\boldsymbol{o}_i^T \boldsymbol{q}_i^\top\right)}{\sum_{j \in \mathcal{B}} \exp\left(\boldsymbol{o}_i^T \boldsymbol{q}_j^\top\right)}, \tag{6}$$

For the finer control at the token level, we propose a contrastive loss to regularize the consistency of the retained components with the reference video. It constrains the distance between the positive retained tokens $\tilde{\boldsymbol{t}}_{R,1}$ and reference video feature $\boldsymbol{p}$ as:

$$\mathcal{L}_R = -\frac{1}{|\mathcal{B}|} \sum_{i \in \mathcal{B}} \log \frac{\exp\left(\tilde{\boldsymbol{t}}_{r,1,i} \boldsymbol{p}_i^\top\right)}{\sum_{j \in \mathcal{B}} \exp\left(\tilde{\boldsymbol{t}}_{r,1,i} \boldsymbol{p}_j^\top\right)}. \tag{7}$$

We also adopt the excluded component to generate negative samples and regularize the negation semantics in the target video. Although there are clean excluded tokens $\tilde{\boldsymbol{t}}_{E,2}$, to avoid the information degradation caused by merging with $\boldsymbol{t}_{R,2}$, we use the initial retained tokens $\tilde{\boldsymbol{t}}_{R,1}$ which contains both retained and excluded components to align with $\boldsymbol{p}$ from the reference database through CMFA. The feature of the identified reference video $\boldsymbol{x}_E^T$ is then fused with $\tilde{\boldsymbol{t}}_I$ in CMFF, obtaining the negative fused video feature $\tilde{\boldsymbol{o}}^T$. We employ a triplet loss in this regularization term to penalize the presence of negation semantics in videos with margin $m$:

$$\mathcal{L}_N = \max(0, m + d(\boldsymbol{q}, \boldsymbol{o}^T) + d(\boldsymbol{q}, \tilde{\boldsymbol{o}}^T)), \tag{8}$$

where $d(\cdot)$ represents the distance metric between the two features.

## 4.3 OVERALL LOSS

The overall loss function is defined as the combination of sentence-level contrastive loss, token-level contrastive loss, consistency regularization term $\mathcal{L}_R$, and negation semantic regularization term $\mathcal{L}_N$:

$$\mathcal{L} = \mathcal{L}^T + \mathcal{L}^S + \mathcal{L}_R + \lambda \mathcal{L}_N, \tag{9}$$

Table 2: Performance on our dataset compared with baseline and existing methods.

| Method | Backbone | R1 | R5 | R10 | R50 |
|---|---|---|---|---|---|
| Video-only | CLIP | 8.27 | 27.89 | 40.69 | 70.00 |
| Text-only | CLIP | 2.86 | 9.19 | 14.61 | 34.15 |
| Video-text sum | CLIP | 8.47 | 24.85 | 46.08 | 65.01 |
| Linear layer | CLIP | 14.68 | 41.42 | 57.82 | 88.74 |
| Linear layer | BLIP | 19.13 | 51.79 | 68.60 | 93.62 |
| TIRG (Vo et al., 2019) | Resnet | 0.00 | 25.80 | 41.20 | 75.92 |
| Artemis (Delmas et al., 2022) | Resnet | 4.76 | 17.81 | 28.39 | 57.92 |
| CosMo (Lee et al., 2021) | Resnet | 7.82 | 25.38 | 37.45 | 70.96 |
| MAAF (Dodds et al., 2020) | CLIP | 2.16 | 20.61 | 33.37 | 70.39 |
| Uncertainty-R (Chen et al., 2022) | CLIP | 5.23 | 18.45 | 28.28 | 60.78 |
| Pic2word (Saito et al., 2023) | CLIP | 8.10 | 25.79 | 38.32 | 71.01 |
| TFR-CVR (Hummel et al., 2024) | CLIP | 15.21 | 40.12 | 52.78 | 81.75 |
| Combiner (Baldrati et al., 2022) | CLIP | 19.57 | 49.46 | 65.38 | 92.11 |
| FreestyleRet (Li et al., 2024) | CLIP | 20.39 | 52.98 | 68.37 | 93.03 |
| CoVR (Ventura et al., 2024) | BLIP | 17.05 | 41.57 | 56.60 | 85.56 |
| FDCA-CLIP | CLIP | 25.84 | 55.84 | 70.23 | 94.33 |
| FDCA-BLIP | BLIP | **26.79** | **63.21** | **78.65** | **97.25** |

where $\lambda$ is the weight of the negation semantic regularization term. With the help of the three additional losses with clear functionalities, the framework can focus on both global and fine-grained information. Our loss design enhances the model's capability to identify relevant information in the reference video, ensuring that the fused video features align more accurately with the modification texts.

## 5 EXPERIMENTS

### 5.1 EXPERIMENTAL SETUP

All the experiments are conducted on FineCVR-1M with recall at $k$ as the evaluation metric. We reproduce several composed retrieval methods including CIR and CVR paradigm. Our image encoder is initialized with both CLIP (Radford et al., 2021) and BLIP (Li et al., 2022) weights. Mean-pooling is used as the temporal encoder in all CIR methods, including our FDCA, to ensure fair comparisons.

### 5.2 COMPARISON RESULTS

Comparison results between our method, the baseline group and Composed Image/Video Retrieval methods group are shown in Table 2.

**Comparison with the pre-trained models.** To demonstrate the capability of pre-trained models for CVR task, we set up several baselines with CLIP feature, including utilizing only the reference video feature, utilizing only modification text feature, summing the video feature and text features directly. The fourth and fifth baselines indicate a linear layer trained for fusing cross-modal features with CLIP and BLIP, respectively. Our method significantly outperforms the above five baseline methods, emphasizing the importance of the combined query. Moreover, the score of video-only baseline surpasses that of the video-text sum baseline, indicating simple feature summation can hardly facilitate the complex CVR task, due to the potential misunderstanding and misrepresentation of the modification texts. In contrast, FDCA focuses on disentangling the modification text regarding the reference video at both sentence and token levels, facilitating more elegant semantic component extraction and alignment in the combined queries.

**Comparison with Existing Methods.** FDCA demonstrates superior performance, thanks to the disentangling of both global and fine-grained information. Notably, approaches such as Artemis underperform the baseline methods because they are not developed based on large-scale pre-trained model. Nevertheless, even with the advanced pre-trained model, it is worth highlighting that our FDCA-CLIP even outperforms the CoVR method fine-tuned with BLIP. This observation indicates that a well-designed feature disentanglement pipeline is crucial in the CVR task, which, in a sense, appears to be more important than updating to a stronger large-scale pre-trained model.

## 5.3 Ablation Studies

We conduct ablation studies to assess the necessity of each component in our model on the FineCVR-1M test set. We use CLIP as the image encoder for the case study, and similar observations can also be obtained on BLIP.

**Ablation Studies on SFD and ATD.** Table 3 shows that FDCA leverages the strengths of both SFD and ATD. The approach without disentangling performs poorly due to misinterpreted combined queries. While introducing SFD helps, it still lacks fine-grained information modeling. The method with ATD excels on R1 by focusing on fine-grained details, but it neglects global information, leading to weaker performance on R5, R10, and R50. By combining ATD and SFD, our model effectively balances the global and fine-grained information, achieving strong results across all metrics. Moreover, within our FDCA framework, we implement CMFA and CMFF separately using distinct Transformers. When we replace these with a unified Transformer, denoted as "w/ Trans.", we find that the alignment-fusion-separate paradigm is more effective than implementing alignment-fusion in a single module. This finding is consistent with previous study Li et al. (2025), and it suggests that aligning the retained information before merging it with the retained features allows the model to better focus on the relevant components.

Table 3: Ablation studies on each module.

| Method | SFD | ATD | R1 | R5 | R10 | R50 |
|---|---|---|---|---|---|---|
| w/o Disent. | | | 22.03 | 53.22 | 69.32 | 94.33 |
| w/ SFD | ✓ | | 22.76 | 53.20 | 68.40 | 94.19 |
| w/ ATD | | ✓ | 25.77 | 50.94 | 64.46 | 90.32 |
| w/ Trans. | ✓ | ✓ | 24.08 | 54.83 | **70.83** | **94.75** |
| FDCA | ✓ | ✓ | **25.84** | **55.84** | 70.23 | 94.33 |

**Ablation Studies on Loss Functions.** In Table 4, each loss function contributes to FDCA, resulting in a well-balanced performance across all metrics. Specifically, the method employing $\mathcal{L}^S$ solely focuses on global sentence-level feature disentangling, showing lower scores at R1. Conversely, the method only utilizing $\mathcal{L}^T$ fails to obtain clean components, leading to the lowest performance. By further introducing the $\mathcal{L}_R$ and $\mathcal{L}_N$, the fusion module can extract fused features that align with the retained and injected components while filtering out the excluded ones, resulting in higher scores. We also implemented a method that uses a single clustering on tokens, which yielded poor results. This emphasizes the importance of clustering clean components. Overall, with the addition of three extra loss functions, FDCA achieves clear disentangled features, resulting in better-fused video representations.

Table 4: Ablation studies on each loss.

| Method | $\mathcal{L}^S$ | $\mathcal{L}^T$ | $\mathcal{L}_R$ | $\mathcal{L}_N$ | R1 | R5 | R10 | R50 |
|---|---|---|---|---|---|---|---|---|
| w/ SFD | ✓ | | | | 22.76 | 53.20 | 68.40 | 94.19 |
| w/ ATD | | ✓ | | | 12.42 | 37.78 | 53.89 | 86.40 |
| w/ SFD+ATD | ✓ | ✓ | | | 22.15 | 52.66 | 68.13 | 93.83 |
| w/ SFD+ATD | | ✓ | ✓ | ✓ | 21.37 | 51.59 | 67.42 | 93.89 |
| w/ SFD+ATD | ✓ | | ✓ | ✓ | 20.83 | 51.56 | 67.35 | 93.85 |
| w/ SFD+ATD | ✓ | ✓ | | ✓ | 24.42 | 53.94 | 68.95 | 93.78 |
| w/ SFD+ATD | ✓ | ✓ | ✓ | | 23.17 | 53.36 | 68.23 | 94.00 |
| w/ single cluster | ✓ | ✓ | ✓ | ✓ | 22.01 | 52.46 | 67.96 | 93.81 |
| FDCA | ✓ | ✓ | ✓ | ✓ | **25.84** | **55.84** | **70.23** | **94.33** |

## 5.4 Analysis

**Effect of Auxiliary Token Disentangling.** In Figure 4, we visualize the cross-modal features obtained through the alignment module using image features from the second last layer of the visual encoder. In Sample 1, we can observe that by integrating ATD, the network significantly enhances the focus on the retained component (red), *i.e.*, the `kitchen`. Moreover, even without the excluded text, the modification text can be distinguishable as retained text (red) and injected text (blue). In

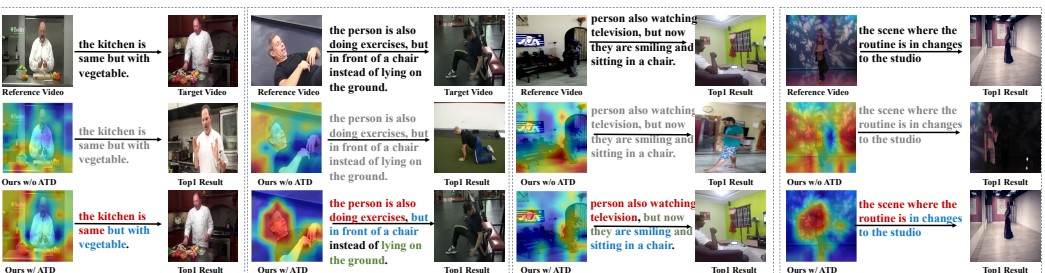

Figure 4: Visualization of FDCA with and without ATD. With the help of the ATD, the model can focus on the retained component in the reference video.

Figure 5: The visualization of failed cases.

Sample 2, when the model lacks ATD, it concentrates wrongly at the excluded component (red), *i.e.*, the `ground`. With the help of ATD, the model shifts its focus to the action of `doing exercises`, demonstrating the essential role of ATD.

**Auxiliary Mechanism or Main Mechanism.** In FDCA, the ATD exclusively participates in the training process. To maximize the utilization of the features from ATD, we validate the performance under the prior-fusion and post-fusion settings. Prior-fusion involves the integration of features before the CMFA and CMFF, while post-fusion incorporates features subsequent to these two modules. The results in Table 5 show that our auxiliary mechanism outperforms these two fusion methods. Notably, the prior-fusion method surpasses the post-fusion. This observation suggests that features at the sentence and token level exhibit a certain level of redundancy, and the subsequent alignment in the prior-fusion effectively mitigates this redundancy. In comparison, our approach directly utilizes the sentence-level feature, thereby circumventing this redundancy issue and achieving superior results.

Table 5: Results on different fusion mechanisms.

| Method | R1 | R5 | R10 | R50 |
|---|---|---|---|---|
| Prior Fusion | 25.70 | **56.03** | **70.35** | 94.29 |
| Post-Fusion | 23.29 | 53.28 | 68.11 | 93.62 |
| FDCA | **25.84** | 55.84 | 70.23 | **94.33** |

**Case Studies.** We analyze the failed case that ranks out of 100 via the FDCA-BLIP to show the limitation of our method and the challenge of the FineCVR-1M dataset. As depicted in Figure 5, our dataset primarily poses a significant challenge for algorithms in the need to balance visual and semantic similarity. In the first case, the retrieved video may closely resemble the reference video but might not clearly show `bread` in the first case, possibly due to the small object scale of the `bread`. In the second case, the algorithm needs to accurately understand the event `sneezing`, but our method finds a video that only shows `sitting` action. This highlights the importance of temporal modeling, which involves capturing the dynamic changes in the video over time and understanding how they relate to the semantic content. Overall, addressing these challenges requires algorithms to further deal with the spurious and missing correlation in the appearance and temporal information in the video with a more elegant design.

**Different Temporal Encoders.** We explore three temporal encoders in Table 6. Following CLIP4Clip (Luo et al., 2022), we replace the temporal encoder with LSTM and Transformer. The results, consistent with CLIP4Clip, show that mean-pooling yields better performance. However, this does not imply that the input video is insignificant, as Table A7 in the Appendix shows that using only images leads to worse results. These findings suggest that a simple temporal encoder is insufficient for fine-grained temporal modeling. More advanced temporal encoders are needed for fine-grained retrieval.

Table 6: Results on different temporal encoders.

| Method | R1 | R5 | R10 | R50 |
|---|---|---|---|---|
| FDCA-MeanP | **25.84** | **55.84** | **70.23** | **94.33** |
| FDCA-Trans | 24.45 | 54.53 | 69.07 | 93.91 |
| FDCA-LSTM | 23.17 | 53.36 | 67.71 | 91.65 |

## 6 CONCLUSION

This paper introduces a new dataset FineCVR-1M to facilitate the study of fine-grained CVR and down-stream adaption of large-scale pre-trained models. Moreover, we propose a method FDCA for fine-grained CVR. FDCA extracts cross-modal fused features by disentangling text features at both the sentence and token levels regarding the reference video. Our experiments demonstrate that the dataset we create has good quality and diversity, and our method achieves remarkable performance compared to other competitors including the original pre-trained VLMs and other CVR methods. Further study includes more intriguing modeling of the complex video temporal information and integration into the multi-modal foundation model development, especially from the perspectives of fine-grained video information seeking and video-centric dialog applications.

## REPRODUCIBILITY STATEMENT

To ensure the reproducibility of our work, we provide comprehensive details on datasets in Section 3 and Section B, method design in Section 4. Details of the training strategies can be found in Section A. The source code has been submitted as supplementary materials.

## ACKNOWLEDGEMENT

This work was supported in part by the National Key R&D Program of China under Grant 2023YFC2508704, and in part by the National Natural Science Foundation of China: 62236008, 62022083, 62306092, and 62441232. The authors would like to thank Shufan Shen, Zhengqi Pei, and the anonymous reviewers for their constructive comments and suggestions that improved this manuscript.

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

# A EXPERIMENTS

## A.1 EXPERIMENTAL SETUP

**Baseline Methods.** To assess the necessity of this CVR paradigm, we evaluate the following baselines by a frozen CLIP:

- Video-only uses the reference video as input only.
- Text-only queries by the modification text.
- Video-text sum is to sum the reference video feature and modification text feature from CLIP directly.
- Linear layer is to employ a linear layer to compose the features of the reference video and modification text from CLIP or BLIP.

**Composed Query Retrieval Methods.** We compare our FDCA with several reproduced composed query retrieval methods including CIR and CVR tasks, as detailed below:

- TIRG (Vo et al., 2019) employs a gated module and a residual module to learn the transformation and preservation features.
- MAAF (Dodds et al., 2020) introduces a self-attention model to enhance the interaction of the cross-modality features.
- CosMo (Lee et al., 2021) uses content and style modulators to learn the underlying style information and residual information.
- Uncertainty-R (Chen et al., 2022) extracts the multi-grained feature and introduces uncertainty regularization to adapt the matching objective.
- Artemis (Delmas et al., 2022) sums the implicit similarity and explicit match scores to produce the final score for retrieval.
- Pic2word (Saito et al., 2023) transforms an input image to a language token, then composes the pseudo token with text tokens through 3 fully-connection layers to obtain the fused feature.
- Combiner (Baldrati et al., 2022) contains 5 linear layers to combine the two modalities feature from the CLIP features.
- FreestyleRet (Li et al., 2024) propose style-space construction and a prompt-tuning strategy structure.
- CoVR (Ventura et al., 2024) finetunes the text encoder of the BLIP model directly.
- TFR-CVR (Hummel et al., 2024) use an LLM to combine the video caption and textual modifier into a coherent target caption.

We reproduce all the compared methods in the CVR paradigm by inputting a video and a text. To encode videos that have an additional temporal dimension, **we utilize mean-pooling in the temporal dimension as the temporal encoder for all the CIR methods and our FDCA**.

**Implementation Details.** All experiments are conducted on the NVIDIA RTX3090 using PyTorch. For our proposed method FDCA, we utilize the frozen CLIP Res50x4 ($d = 640$) (Radford et al., 2021) or BLIP large ($d = 256$) (Li et al., 2022) as our video encoder. The model is optimized with Adam with an initial learning rate of 1e-4. We set the batch size to 1024 to maintain the performance. To avoid overfitting, we train our FDCA for 30 epochs. The $m$ in the negation semantic regularization term $\mathcal{L}_N$ is 0.2, while the weight $\lambda$ of the negation semantic regularization term $\mathcal{L}_N$ is set as 5. We implement the Cross-Modality Feature Alignment (CMFA) and Cross-Modality Feature Fusion (CMFF) modules using six Transformer Encoder layers. Each encoder layer consists of a multi-head attention mechanism, a Layer Normalization, and a Feed-Forward Network.

Table A7: Results of Different Query for Video Retrieval on FineCVR-1M test dataset.

| Train/Val | Test | R1 | R5 | R10 | R50 |
|---|---|---|---|---|---|
| img+mod | img+mod | 20.51 | 43.19 | 60.71 | 90.12 |
| | vdo+mod | 22.18 | 45.82 | 63.92 | 91.80 |
| vdo+mod | img+mod | 15.00 | 37.10 | 49.32 | 76.88 |
| | vdo+mod | **25.84** | **55.84** | **70.23** | **94.33** |

Table A8: Results on WebVid-CoVR test dataset.

| Method | Finetuned BLIP's text encoder | R1 | R5 | R10 | R50 |
|---|---|---|---|---|---|
| CoVR | ✓ | 53.13 | 79.93 | 86.85 | 97.69 |
| FDCA-BLIP | | 52.23 | 79.42 | 86.66 | 96.91 |
| FDCA-BLIP | ✓ | **54.80** | **82.27** | **89.84** | **97.70** |

## A.2 RESULTS ON DIFFERENT QUERY FOR VIDEO RETRIEVAL

To explore the relative effectiveness of images and videos in the CVR task, we conduct validations on the FineCVR-1M test dataset using queries that included modification text either composed with an image (img+mod) or a video (vdo+mod). The image is the middle frame of the video. As shown in Table A7, due to the rich action descriptions in our dataset, whether trained with image+mod or video+mod, the test results for image+mod are comparatively lower. This indicates that the visual information provided by a single image is insufficient to convey complete semantic information. In contrast, videos can not only provide more comprehensive visual information but also offer fine-grained contextual information, making the performance of vdo+mod better.

## A.3 RESULTS ON OTHER CVR DATASETS

**WebVid-CoVR-Test Dataset**: We employ the CoVR and FDCA-BLIP for training and testing on the WebVid-CoVR dataset, and the results are shown in Table A8. The results for CoVR are from the paper (Ventura et al., 2024). It can be observed that our method is slightly inferior to directly fine-tuned CoVR. This may be due to the fact that modification texts in the WebVid-CoVR dataset are often incomplete sentences, which hinders the provision of the three essential components, consequently impacting the effectiveness of the FDCA method. Additionally, CoVR employs the fine-tuned BLIP directly, while our FDCA only utilizes the frozen BLIP. This may be another reason why ours is slightly inferior to CoVR. Therefore, we adjust our training strategy by reducing the weight $\lambda = 1$ of $L_T$, setting $m = 0.05$, and fine-tuning BLIP's text encoder following CoVR's configuration. These modifications led to improved performance.

**EgoCVR Test Dataset**: We also validate our method on the EgoCVR dataset (Hummel et al., 2024). Due to the unavailability of training data, we directly apply our FDCA-BLIP model pretrained on FineCVR-1M for testing. The results are presented in Table A9. TFR-CVR (Hummel et al., 2024) achieves superior performance as it leverages pre-trained video-language weights from the EgoCVR dataset and requires no additional training. In contrast, both our method and CoVR require training but lack access to the training set, resulting in relatively lower performance. Nevertheless, our approach still outperforms CoVR, particularly on the fine-grained R1 metric, demonstrating the effectiveness of our decomposition strategy in fine-grained retrieval tasks.

## A.4 PARAMETER SENSITIVITY

In the ATD, we leverage the negation semantic regularization term $\mathcal{L}_N$ to penalize the presence of negation semantics in videos. We study the effects of two parameters including $m$ and $\lambda$ in the negation semantic regularization term $\mathcal{L}_N$ on the FineCVR-1M validation set.

**Effect of $m$.** $m$ is the margin value between positive samples and negative samples. As shown in Figure A6a, while the performance variations across different margins $m$ are small in our FineCVR-

Table A9: Results on Ego-CVR test dataset.

| Method | Global | | | Local | | |
|---|---|---|---|---|---|---|
| | R1 | R5 | R10 | R1 | R5 | R10 |
| CoVR (Ventura et al., 2024) | 5.4 | 15.2 | 24.3 | 33.1 | 49.5 | 62.9 |
| FDCA-BLIP | 8.7 | 19.2 | 27.1 | 36.3 | 51.1 | 63.8 |
| TFR-CVR (Hummel et al., 2024) | **14.1** | **39.5** | **54.4** | **44.2** | **61.0** | **73.2** |

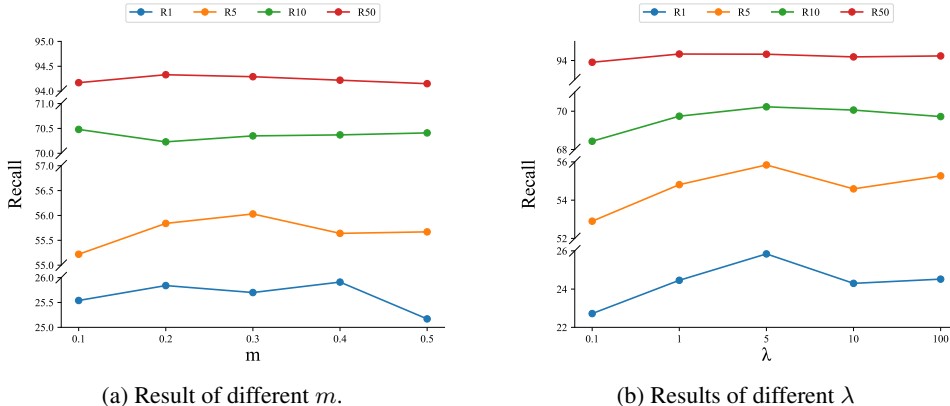

(a) Result of different $m$.  (b) Results of different $\lambda$

Figure A6: The analysis of parameters in the negation semantic regularization term. (a) Effect of $m$. (b) Effect of $\lambda$

1M dataset, there is a decline in performance as $m$ increases. The best performance's $m$ is located at $m = 0.2$, which is used in our method. This underscores the effectiveness of our ATD in generating negative samples to filter misleading information.

**Effect of $\lambda$.** $\lambda$ is the weight of negation semantic regularization term $\mathcal{L}_{\mathrm{N}}$. From Figure A6b, the results demonstrate an initial upward trend followed by a subsequent decline. We choose a parameter that yields consistently good overall performance, with $\lambda = 5$.

## A.5 THE IMPORTANCE OF THREE COMPONENTS

The FDCA with ATD only can decompose the token into three components including retained, injected, and excluded components. Hence, to explore the importance of the three components, we ablate each component in the final fusion module. Table A10 shows that all three components are essential. The performance with retained components is not bad, since the retained components can make the model focus on the target video which is similar to the retained video feature. Moreover, when the injected component is introduced, the model can produce a fused video feature that meets the modified needs of the user. Furthermore, with the help of the excluded component, the model can filter the results that the user doesn't want. All these components help the model understand the visual and semantic demands of users clearly.

Table A10: Performance on three components by ATD.

| Retained | Injected | Excluded | R1 | R5 | R10 | R50 |
|---|---|---|---|---|---|---|
| ✓ | | | 18.20 | 42.98 | 60.12 | 89.10 |
| ✓ | ✓ | | 24.15 | 53.29 | 68.91 | 92.96 |
| ✓ | ✓ | ✓ | **25.84** | **55.84** | **70.23** | **94.33** |

Table A11: Results on better temporal encoding mechanism.

| Method | R1 | R5 | R10 | R50 |
|---|---|---|---|---|
| FDCA-Trans | 24.45 | 54.53 | 69.07 | 93.91 |
| FDCA-Trans-Dicosa (Jin et al., 2023b) | 25.18 | 51.12 | 64.07 | 90.25 |
| FDCA-Trans-Dicosa+retained | **25.79** | **55.90** | **69.94** | **94.05** |

Table A12: Training time comparison.

| Method | Parms | Processed data | Epoch | Training time per epoch | Total time |
|---|---|---|---|---|---|
| CoVR (Ventura et al., 2024) | 446M | Image | 5 | 30 h | 150 h |
| Pic2word (Saito et al., 2023) | 179M | CLIP feature | 30 | 6 min | 3 h |
| Combiner (Baldrati et al., 2022) | 237M | CLIP feature | 100 | 10 min | 16 h |
| FDCA | 570M | CLIP feature | 30 | 20 min | 10 h |

## A.6 EXPLORATION OF BETTER TEMPORAL ENCODER

Intuitively, since video composed retrieval requires understanding contextual information within videos, incorporating a superior temporal encoder would yield better results. Here, we employ the temporal encoding mechanism in DiCoSA (Jin et al., 2023b) in Table A11. Its effectiveness is limited to R1 metrics. We believe this limitation arises because DiCoSA's text information, which directly corresponds to video content, can be aligned with videos for weighted sum computation. In contrast, our text information contains modification-related semantics, and using weighted sums introduced certain biases. When we instead use our decomposed residual features for weighted sum computation, we observed significant performance improvements. This validates both the effectiveness of our decomposition approach and demonstrates that superior temporal encoders can enhance temporal understanding.

## A.7 COMPARISON OF TRAINING AND INFERENCE PERFORMANCE.

We compare training time in Table A12, and inference time in Table A13. Although our model has the most parameters, it converges faster and has a faster inference speed. Since we only use SFD during the inference stage, the inference speed is only slightly slower than the Combiner (Baldrati et al., 2022).

## A.8 VISUALIZATION OF ATD

We offer some illustrative examples of ATD results in Figure A7. As observed, ATD can effectively disentangle the modification text into three distinct components: clean retained tokens, injected tokens, and excluded tokens. Because we guide the disentangling of initial retained tokens through reference video features, the tokens are typically crucial retained components in the reference video, such as concepts like `..bathroom.. but rearranging on the sink` in the first example and `.. singing, but..` in the seventh example. On the other hand, the injected tokens are the remaining parts of the modification text. Therefore, injected tokens usually include concepts from the target video as well as grammatical elements, such as `and the person is also in ..  the takes a pill ..  objects` in the first example and `the ..  is`

Table A13: Inference time comparison.

| Method | Parms | Processed data | Inference time per |
|---|---|---|---|
| CoVR (Ventura et al., 2024) | 446M | Image | 56 pairs/s |
| Pic2word (Saito et al., 2023) | 179M | Image | 149 pairs/s |
| Combiner (Baldrati et al., 2022) | 237M | Image | 107 pairs/s |
| FDCA | 570M | Image | 84 pairs/s |

`..` `with net` in the sixth example. After the second clustering, the initial retained token can be disentangled into clean retained tokens `bathroom` and excluded tokens `but rearranging on the sink`. Notably, even in the modification text without negation words, the ATD can also detect the excluded tokens such as the `boy` that should not exist in the target video in the fourth example.

| Reference Video & Target Video | Mod Text | Initial Retained Positive Tokens | Injected (Positive) Tokens | Initial Retained Tokens | Clean Retained (Positive) Tokens | (Clean) Excluded Tokens | Excluded Tokens w/ Negation Words |
|---|---|---|---|---|---|---|---|
| | the person is also in the bathroom, but takes a pill instead of rearranging objects on the sink. | .. bathroom .. but rearranging on the sink | the person is also in .. the takes a pill .. objects | bathroom .. but instead of rearranging on the sink | ..bathroom.. | but rearranging on the sink | but instead of rearranging on the sink |
| | the home office / study is same but with laptop | office study .. same but .. | the home .. is .. with laptop | office study .. same but .. | .. office study .. same but .. | - | - |
| | the Living room is same but with a broom | the living room .. same but .. | is .. with a broom .. | the living room .. same but .. | the living room .. same but .. | - | - |
| | the person changes from a boy to a woman | .. boy to a .. | the person changes from a .. woman | .. boy to a .. | .. to a .. | .. boy .. | .. boy .. |
| | the Home Office / Study is same but with chair | .. office / study .. same but .. | the home .. is .. with chair | office / study .. same but .. | office / study .. same | .. but .. | .. but .. |
| | the arena is same but with net | .. arena .. same but .. | the .. is .. with net | .. arena.. but | .. arena same.. | ..but.. | ..but.. |
| | the person is also singing, but on the show with a good judge | .. singing , but .. | the person is also .. on the show with a good judge | .. singing , but.. | ..singing.. | ..but.. | ..but.. |
| | the scene where the vacuum is in changes to the Kitchen | .. vacuum is in changes to .. | the scene where the.. the kitchen | .. vacuum is in changes to .. | vacuum is .. to .. | .. in changes .. | .. in changes .. |
| | the person takes another shot and makes it and we see a recap of that as well | .. shot and .. and .. that.. | the person takes another .. makes it .. we see a recap of .. as well | .. shot and .. and .. that .. | .. shot and .. and .. as.. | .. we .. | .. we .. |
| | the person is also spreading something on bread, but it has changed to butter instead of mustard | .. spreading .. on bread .. but .. to .. | the person is also .. something .. it has changed .. butter | .. spreading .. on bread , but .. to .. instead of mustard | .. spreading .. on .. but .. to mustard | .. instead of bread .. | .. bread .. |

Figure A7: Examples of ATD results.

# B  THE FINECVR-1M DATASET

As shown in Figure A8, we first match similar videos as the reference video and the target video, respectively. We then automatically generate modification texts regarding static concepts and action concepts, respectively. The former generates modification texts by filling the different key static concepts into the textual prompts, while the latter leverages a fine-tuned LLM to generate the action difference description directly.

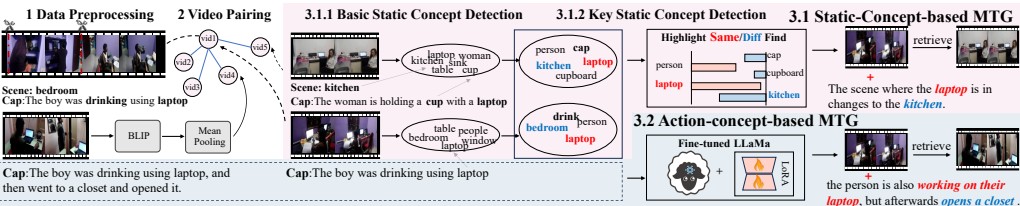

Figure A8: Pipeline of dataset construction. We segment long videos into video clips by preprocessing and pairing video clips by similarity comparison. Then we use prompts and LLM to generate modification texts regarding static concepts and action concepts, respectively.

## B.1  VIDEO PREPROCESS AND PAIRING.

We begin by extracting videos from public datasets and clipping them based on event timestamps provided in the annotations. Videos without annotations are excluded from the dataset.

We establish video pairs from the same video source by utilizing cosine similarity, a widely used metric in video similarity learning (Kordopatis-Zilos et al., 2019; Xu & Wang, 2021; Chen et al., 2020a). Specifically, we uniformly sample 8 frames from each video and compute their frame-level features using BLIP-2 (Li et al., 2023). These features are then mean-pooled to obtain a compact video-level representation. Using cosine similarity between compact video features to compute video similarity, we identify the top 20 most similar videos for each query video, forming pairs that serve as the foundation for subsequent text modification text generation. To further ensure dataset quality, we instruct annotators to exclude irrelevant video pairs during test set construction.

## B.2  MODIFICATION TEXT GENERATION

**For static concepts**, we use fixed templates to generate differences and similarities among key concepts. To ensure the high quality of the static concepts, concepts in our FineCVR-1M dataset are obtained from the annotations and captions of four accessible benchmark datasets (Ji et al., 2020), as well as high-confidence score results by BLIP-2. Figure A9 illustrates the detailed process of key object calculation. In detail, objects in the caption are more semantically meaningful for humans and represent key concepts, thus receiving higher scores of 1.0. Object annotation is also critical for this task, so we give it 0.5. We believe that the results from BLIP-2 provide basic visual information with low confidence, so we assign a score of 0.1 to the BLIP result. Moreover, the importance of words in the caption is related to their location, with the last word being less important than the first word. Hence, we assign the first object in the caption list as the highest location score, and the last object a location score of 0. In summary, our FineCVR-1M dataset benefits from the utilization of these four reliable sources of data, resulting in a dataset with high-quality key concepts.

Then we compare key static concepts between video pairs and generate descriptions of their differences with three types of prompts: **(a)** identify the scene difference where the same object appears in the video pair like *"the scene where the {***} is in changes to the {***}"*; **(b)** choose a different object as the focus of the prompt for videos with the same scene, formed as *"the {***} is same but with {***}"*; and **(c)** replace the attributes of the object in the target video when the key object concept is the same, *e.g.*, *"the attribute of the {***} is replaced by {***}"*.

Furthermore, to explore the impact of fixed templates, we also rewrite 6,043 sample texts using GPT-4 for the test set. Table A14 shows that flexible texts have a minimal impact on results, since the

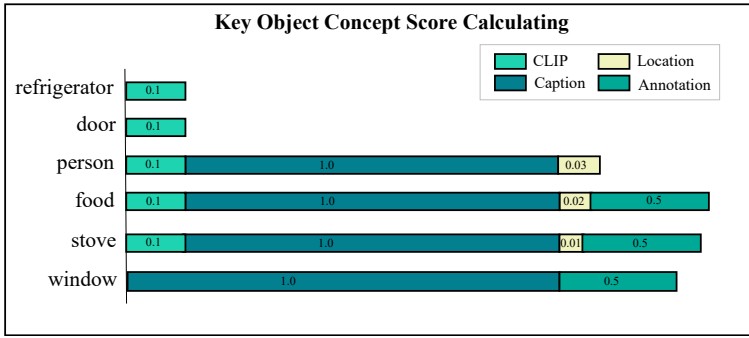

Figure A9: A sample of calculating key object scores. The caption is "a person cooks food on a stove before looking out of a window", and the CLIP detects that there are [refrigerator, door, person, food] in this video. The objects in the caption are [person, food, stove, window], and the objects in the annotation are [food, stove, window]. After calculating, the key object is "food" with the highest score of 1.62.

Table A14: Test result on rewritten flexible modification text by GPT-4.

| Method | Dataset | R1 | R5 | R10 | R50 |
|---|---|---|---|---|---|
| FDCA-CLIP | FineCVR-1M | 14.69 | 46.17 | 64.87 | 94.44 |
| | Rewritten | 14.23 | 43.31 | 61.91 | 92.45 |
| FDCA-BLIP | FineCVR-1M | **18.68** | **52.39** | **70.99** | **95.28** |
| | Rewritten | 18.08 | 51.32 | 70.23 | 94.89 |

CLIP or BLIP text encoder is robust enough to comprehend the core semantic meaning. Moreover, the gap between the FDCA-CLIP is bigger than that between the FDCA-BLIP, which could be attributed to the BLIP encoder being BERT, capable of handling complex sentences. This indicates that for a superior text encoder, the flexibility of the text will have little impact on the results as long as the core content remains consistent.

**For action concepts**, we use LLM to generate differences and similarities between video caption pairs. We first attempt to use ChatGPT (Ouyang et al., 2022) to generate 1,000 data samples for validation. Specifically, we input the following prompt with caption pairs into ChatGPT to generate 1000 samples into **(d)** different actions with similarities and **(e)** different actions:

> I'll give two ordered video captions, and you should compare two videos and complete a change captioning task based on something similar and return to me with a concluded sentence as the following template. If the same action or object or event exists in both two ordered videos, you should fill this same thing in the first {} and output: "the person is also {} in both two videos, but {} in the second video ". Else you output: "The person changes to {} in the second video". (please check carefully, if you use the first format, there must be something that exists in both videos). Only give me an output as one of the above formats.

However, we find that the accuracy is not high (below 60%) since ChatGPT has not previously encountered this type of task. To ensure that the LLM could effectively adapt to identifying differences and similarities between video caption pairs, we manually revise the responses of the 1,000 samples. We then use 606 samples as a training set and 394 as a validation set to fine-tune the open-source model LLaMA 2 (Touvron et al., 2023b). As a result, we achieve an accuracy of 92% and use this fine-tuned model to generate the action concepts text.

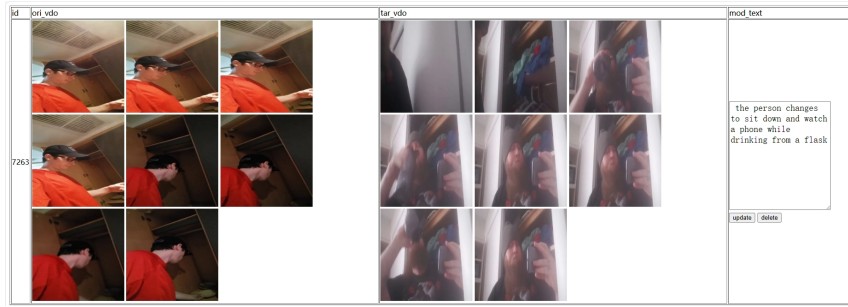

Figure A10: We manually select samples that have visual differences and similarities.

Table A15: The statistics of the prompt class in the dataset.

| ClsID | Method | Prompt | Coverage | Example |
|---|---|---|---|---|
| (a) | Static-Concept | *the scene where the {***} is in changes to the {***}* | 28.54% | the scene where the *bag* is in changes to the *entryway* |
| (b) | | *the {***} is same but with {***}* | 22.08% | the *bedroom* is same but with *screen* |
| (c) | | *the attribute of the {***} is replaced by {***}* | 0.48% | the attribute of the *blanket* is replaced by *throw* |
| (d) | Action-Concept | I'll give two video captions, and you should compare theses | 11.70% | the person is *cooking at the stove*, but *looks for something inside frige* |
| (e) | | and complete change captioning based on something same | 37.20% | the person changes to *pick up clothes instead of shaking a blanket out* |

Additionally, we observe that despite instructing the LLM to generate two categories: **(d)** and **(e)**, it still tends to hallucinate similarities that do not actually exist. To address this, we filter out any triplets where the verbs in the identified similarities were not present in the original video captions.

**For the test set**, we manually select samples on the website as shown in Figure A10 that visually exhibit significant differences, ensuring they are suitable for evaluation. A user group of eight individuals, including some of the authors, is employed to ensure that the generated texts accurately reflect changes between the videos. We also implement a two-round cross-validation process. After completing the manual correction process, we obtain 10,043 triplets as our test set.

### B.3 THE PROPERTIES OF FINECVR-1M DATASET

We summarize two properties and challenges of our FineCVR-1M as follows.

**Variable Concepts.** As mentioned in the main paper, we focus on key concepts in videos in the FineCVR-1M dataset construction. Our FineCVR-1M dataset covers four different concept categories, *i.e.* objects, scenes, attributes, and actions. Comprising five types of prompts, the FineCVR-1M dataset's class coverage is detailed in Table A15. Notably, data from Static-Concept-based MTG and Action-Concept-based MTG roughly account for half of the entire dataset. This indicates that our dataset not only encompasses visual concepts but also integrates rich temporal information. For each class, we also provide a word cloud in Figure A11. Specifically, in the (a) and (b) classes, our FineCVR-1M dataset focuses on common outdoor or indoor scenes such as `sky`, along with nouns representing objects `food`, or elements associated with these scenes `window`. In the (c) class, people pay more attention to the color of objects, such as `brown hair`, and at the same time, some ongoing actions also become specific attributes of objects, such as `connecting computer`. Meanwhile, in the (d) and (e) classes, all the words revolve around human and object actions, such as `playing ball`, showcasing the diversity of actions.

**Fine-grained Demand Semantic.** Compared to the incomplete sentences in existing composed query datasets (Vo et al., 2019; Han et al., 2017; Liu et al., 2021; Wu et al., 2021; Ventura et al., 2024), the modification texts in our FineCVR-1M dataset are all complete sentences. Figure A12a indicates that the length of the modification text in most cases is 11 words, which is longer than the 5 words observed in the WebVid-CoVR dataset (Ventura et al., 2024). As shown in Figure A12b, our dataset covers nouns, verbs, adjectives, and even adverbs, leading the modification text to represent the complete preference of users. Moreover, we statistics the portion of three components

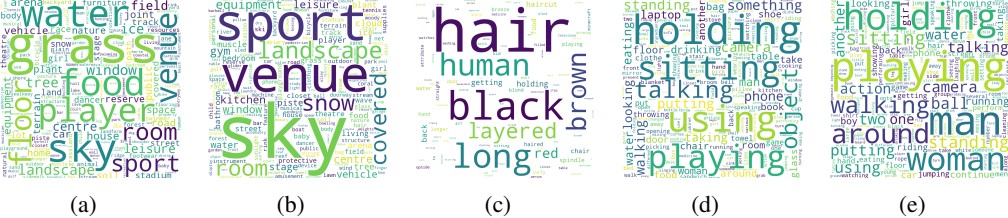

| (a) | (b) | (c) | (d) | (e) |

Figure A11: The word cloud of each class in our dataset.

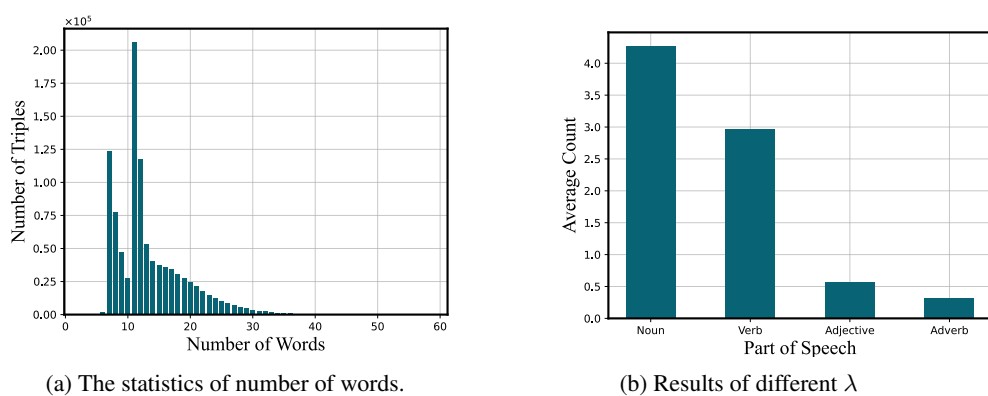

(a) The statistics of number of words.

(b) Results of different $\lambda$

Figure A12: The statistics of modification text. (a) We count the number of words for each modification text. Most modification texts have between 7 and 13 words. (b) We calculate the number of the part of speech in each modification text.

including retained, injected, and excluded components. In FineCVR-1M, 100% modification texts comprise retained components and injected components, and 11.67% texts comprise excluded components. While the abbreviated text in Webvid-CoVR only comprises 77.68%, 100%, and 2.80%, respectively. Due to the rich information in our video, such as multiple individuals, complete semantics and rich components can prevent ambiguity caused by a lack of subject and avoid returning inaccurate video results.

**The Challenges.** The main challenge of our FineCVR-1M dataset is threefold: extracting concepts from videos exhaustively, understanding the semantics of modification text accurately, and fusing the cross-modal feature adequately. Videos from our FineCVR-1M not only involve static concepts but also encompass actions. Therefore, addressing the first challenge requires CVR algorithms to employ an enhanced visual encoder. The visual encoder is capable of effectively detecting concepts, even for small-scale objects and attributes. Additionally, a robust temporal encoder is essential to comprehend the semantic information conveyed by the videos. The second challenge requires algorithms with a powerful text encoder to comprehend modification text, including extracting negation meanings and capturing fine-grained words that represent users' demands. The third challenge involves developing algorithms that can integrate valuable information from both videos and modification texts. These algorithms should discern which parts in reference videos to preserve and which parts to transform, ensuring an optimal fusion of cross-modal features.

### B.4 MORE SAMPLES IN FINECVR-1M

We present some examples from our FineCVR-1M dataset in Figure A13. Thanks to the combined insights from BLIP-2 results, annotations, and fine-tuned LLM, our FineCVR-1M dataset offers a rich blend of concepts including static concepts and actions, showcasing elements like `stairs` and `personal protective equipment`. Additionally, the video captions provide fine-grained

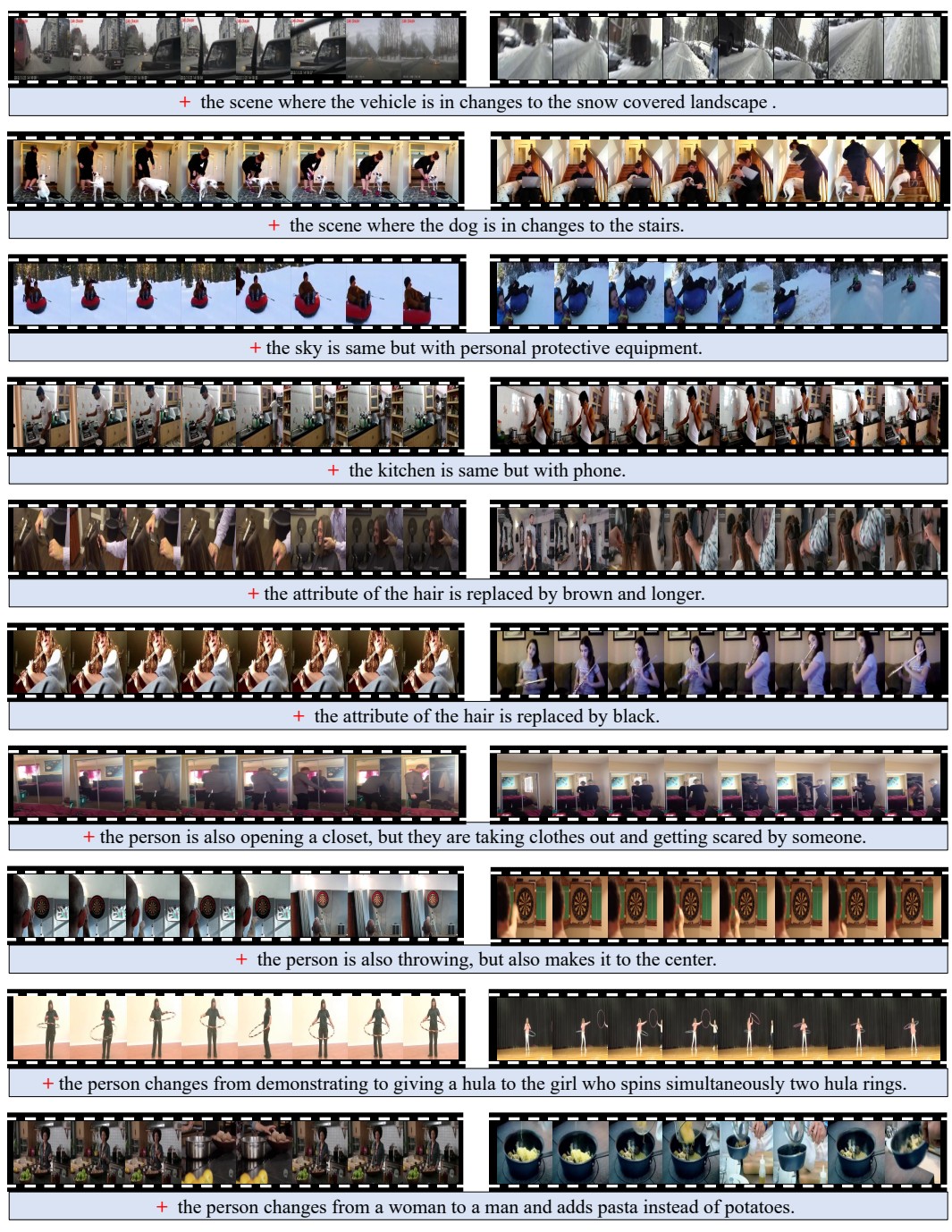

Figure A13: Example tuples in FineCVR-1M dataset. The videos on the left represent reference videos, while those on the right represent target videos. The modification texts at the bottom highlight the differences between the two videos.

information on action changes, capturing details like `giving a hula` and `spins two hula rings`.

