# OpenReview forum: "Learning Fine-Grained Representations through Textual Token Disentanglement in Composed Video Retrieval"
_ICLR.cc/2025/Conference — ICLR 2025 Poster_

### Official Review · Reviewer_JTbd · 2024-10-24

**Soundness:** 2
**Presentation:** 3
**Contribution:** 2
**Rating:** 5
**Confidence:** 5

**Summary:**

The authors introduce FineCVR-1M, a large-scale dataset comprising 1,010,071 video-text triplets generated through an automated process that identifies key concept changes between video pairs. This dataset includes textual descriptions of both static and dynamic concepts, facilitating nuanced video retrieval. The authors further propose the Textual Feature Disentanglement and Cross-modal Alignment (FDCA) framework, which disentangles features at both the sentence and token levels.

**Strengths:**

1. Originality
The introduction of the FineCVR-1M dataset represents a significant advancement in the field of composed video retrieval (CVR). By providing a large-scale dataset with 1,010,071 video-text triplets, the authors address a critical gap in fine-grained video retrieval research.
2. Clarity
The paper is well-structured and clearly communicates its objectives, methodologies, and findings.
3. Significance
The work has impressive implications for the field of video retrieval, particularly in enabling fine-grained retrieval tasks that meet user-specific requirements. Experimental results indicate that the framework effectively extracts and aligns features, achieving superior performance compared to existing methods.

**Weaknesses:**

1. The comparative analysis with other methods is insufficient. In Section 5.2, Table 2 is presented but not mentioned or discussed in the text. This oversight limits the effectiveness of the comparison and fails to guide readers through the results.
2. Are all the experiments conducted solely on the introduced FineCVR-1M dataset? If so, this raises concerns about the generalizability and robustness of the proposed method. Relying on a single dataset may limit the findings and their applicability to other contexts.
3. Although the methods proposed by the authors demonstrate superiority over WebVid and EgoCVR in many aspects, it is essential to conduct experiments on existing datasets to further validate the effectiveness of FDCA. Testing on established benchmarks is crucial for establishing credibility and ensuring that the proposed method generalizes well to different scenarios.
4. Additionally, the results presented in Table A4 indicate that the authors' proposed method performs worse than CoVR in certain metrics. This raises questions about the robustness of the proposed approach.
5. To facilitate the authors' reading experience, it is recommended that all images be provided as vector graphics whenever possible. This approach will help prevent distortion when magnifying figures, ensuring that details remain clear and crisp. Specific figures that would benefit from this adjustment include Figures A10, A12, A14, and A15. Additionally, it has been noted that some of the fonts in Figures 2 and 3 are vector-based while others are scalar. This inconsistency can lead to variations in clarity and readability.
6. Based on the proposed Auxiliary Loss Construction method, there appears to be some degree of redundancy among the four losses defined in Eq. (8), particularly between L^T, L^S, and L^R.
7. In addition to retrieval performance, inference efficiency and model parameters are critical factors in the field of retrieval. The ability to deploy models effectively in real-world applications often hinges on these aspects. For example, Pic2word, which contains only one MLP as trainable parameters, highlights how simplicity can lead to enhanced efficiency without compromising performance.

**Questions:**

1. Do the weights in Eq. 8 have different effects on the results?
2. The specific meaning of L^T and L^S need to be explained.
2. Please refer to the Weaknesses for the other questions.

---

### Official Review · Reviewer_GX4t · 2024-10-25

**Soundness:** 3
**Presentation:** 2
**Contribution:** 3
**Rating:** 8
**Confidence:** 5

**Summary:**

The study introduces the FineCVR-1M dataset and proposes a Feature Disentanglement and Cross-modal Alignment (FDCA) framework, which enhances retrieval by disentangling and aligning text and video features at both the sentence and token levels, achieving superior results in composed video retrieval. I hold a positive attitude toward this research; however, there are a few minor issues in the paper that need correction.

**Strengths:**

The study introduces the FineCVR-1M dataset and proposes a Feature Disentanglement and Cross-modal Alignment (FDCA) framework, which enhances retrieval by disentangling and aligning text and video features at both the sentence and token levels, achieving superior results in composed video retrieval. I hold a positive attitude toward this research.

**Weaknesses:**

1、In line 269, the structure of the Transformer encoder could be explained in the supplementary materials, or a reference could be added.
2、In the Methods section, there is no explanation of how $\mathcal{L}^{T}$ and $\mathcal{L}^{S}$ are derived; please provide this information in the Methods section.
3、In line 432, the title above Table 4 is incorrect; it is labeled as Figure 4. (Table 5; Table 7; Table 9). Please carefully review the manuscript for existing minor errors.
Note: Please revise the Method section carefully, as it is crucial to the paper. Some implementation details that cannot be included in the main text can be added to the supplementary materials, which will help readers understand the underlying principles. I hold a positive attitude toward your work, but the manuscript requires thorough revision.

**Questions:**

1、In lines 253 and 256, the image encoder and text encoder use CLIP; the authors should provide a reference for CLIP.
2、In line 405, please add references after mentioning CLIP and BLIP.

---

> ### Comment · Reviewer_GX4t · 2024-11-26
>
> I have no other suggestions, so I accept the paper. Keep up the good work. I'll keep an eye on your research. Good luck to you!

---

### Official Review · Reviewer_FBRb · 2024-11-01

**Soundness:** 3
**Presentation:** 3
**Contribution:** 3
**Rating:** 8
**Confidence:** 5

**Summary:**

Video Retrieval is a very challenging task, hence the proposal of composite video retrieval, which uses images and text as retrieval signals to search for video content. Current content retrieval faces two issues:
(1). A lack of video-retrieval datasets with fine-grained descriptions.
(2). A lack of effective solutions to implement good video-retrieval.
In response to the first issue, the authors proposed a fine-grained large-scale video retrieval dataset, FineCVR-1M, which includes one million video-text pairs. Addressing the second model-side problem, the authors proposed a decoupled text representation-cross modality alignment model. However, the methodology part requires more ablations and the related works are incomplete. It would be better to address these issues for further ranking improvement.

**Strengths:**

1. The authors have provided a high-quality, fine-grained dataset in the field of video retrieval, which has made significant contributions to the community.

2. The authors analyzed the advantages of this new dataset over the WebVid-CoVR dataset, which is also proposed for the composed video retrieval task. CoVR uses LLM to describe the differences between two videos and does not support fine-grained details. What we generate are fine-grained text descriptions.

3. The authors' dataset structure is quite ingenious, as it introduces LLM to label the three important components between two video pairs: retained component, injected component, and excluded component, to generate fine-grained text descriptions.

**Weaknesses:**

1. In the abstract section, the challenges of this composed video retrieval task are described too simplistically. The difficulty of fine-grained video-text modeling is a long-standing and intractable problem. The authors could provide a more nuanced description of the challenges.

2. The author should conduct a more detailed exploration of the temporal encoder. Dicosa[1] found that using the text-side [cls] token to perform cosine similarity with each video frame feature, and then passing through softmax to convert into a probability distribution to guide the weighted summation of video features is a better compression method. Could this process bring performance improvements in composed video retrieval?

3. The author lacks citations of relevant papers. Specifically, DPC-KNN has also been used in other video retrieval works [2], and relevant studies include Dicosa [2] and FreestyleRet [3]. FreestyleRet contains the exploration of composed image retrieval.

[1]. Text-video retrieval with Disentangled Conceptualization and Set-to-Set Alignment
[2]. Video-Text as Game Players: Hierarchical Banzhaf Interaction for Cross-Modal Representation Learning
[3]. FreestyleRet: Retrieving Images from Style-Diversified Queries

**Questions:**

1. During the construction of the WebVid-CoVR dataset, did the authors consider the possibility of retrieval mismatch due to video length when they mixed short videos like MSRVTT with longer ones like ActivityNet?

---

### Official Review · Reviewer_tW2k · 2024-11-03

**Soundness:** 2
**Presentation:** 2
**Contribution:** 2
**Rating:** 3
**Confidence:** 4

**Summary:**

This paper proposes the FineCVR-1M benchmark, which supports the combined query with both reference videos and modification text for fine-grained video retrieval. The authors also propose FDCA which performs text feature disentangling at sentence and token levels to
progressively enhance the descriptive power of features of the reference video, facilitating efficient retrieval of target videos that visually and semantically satisfy user expectations.

**Strengths:**

1. It is essential to construct fine-grained composed video retrieval datasets and develop the corresponding method. The proposed methodology including collecting similar videos and prompting LLMs for annotation generation is promissing.

2. The proposed FDCA method follows the conventional attention based principle, which has been demonstrated effiective across various tasks.

**Weaknesses:**

1. Unclear dataset construction details: There remain lots of missing details in the dataset construction. For example, how to handle the video when prompting, such as how many frames are used.

2. Quality control of the proposed dataset. In L160, the authors select the top 20 similar videos for each video based on the cosine similarity between them. It is unclear about the cosine similarity selection accuracy. The authors should clarify this through human mannual check or some other solutions.

3. The proposed method is similari to existing method [1]. These two papers share the similar fine-grained cross-modal alignment and fusion methodology. The authros should cite this paper and discuss the differences.

4. In experiments, the authors only include the results in the proposed dataset. It is unclear about the performance of the proposed method in general video-text retrieval datasets. It is recommended to include the discussion about it.

5. The overall paper writing can be polished. For example, the presented figures should be presented as vector images

[1] Disentangled Representation Learning for Text-Video Retrieval

**Questions:**

Please refer to the weakness part.

---

### Meta-Review · Area_Chair_exDd · 2024-12-16

**Metareview:**

This paper introduces the FineCVR-1M benchmark, which supports combined queries with both reference videos and modification text for fine-grained video retrieval. It also proposes the FDCA method for this task.

The paper received divergent ratings (8, 8, 5, 3). Reviewers acknowledged the value of the constructed dataset and the good performance of the proposed FDCA method. However, key concerns remained regarding the lack of dataset details and quality control (reviewer tW2k), the similarity of the method to prior work (reviewer tW2k), and the absence of evaluations on other datasets (reviewers tW2k, JTbd). The rebuttal provided substantial responses to these concerns.

During the post-rebuttal discussion, reviewer tW2k maintained the reservations on these aspects.  The AC partially disagreed with these concerns, offering the following clarifications:
- As mentioned by the other three reviewers, the dataset is of high value and the authors have provided more details in the rebuttal.
- Although the method shares conceptual similarities with prior work, the task and approach exhibit distinct innovations.
- The FDCA method was designed for composed retrieval tasks, and its performance on general benchmarks, while slightly below SOTA, remains competitive.

Given these considerations, as well as the paper's contributions to both the benchmark and method, the AC recommends acceptance. The authors should consider reviewers’ comments and revise the final version accordingly.

**Additional Comments On Reviewer Discussion:**

The paper received divergent ratings (8, 8, 5, 3). Reviewers acknowledged the value of the constructed dataset and the good performance of the proposed FDCA method. However, key concerns remained regarding the lack of dataset details and quality control (reviewer tW2k), the similarity of the method to prior work (reviewer tW2k), and the absence of evaluations on other datasets (reviewers tW2k, JTbd). The rebuttal provided substantial responses to these concerns.

During the post-rebuttal discussion, reviewer tW2k maintained the reservations on these aspects.  The AC partially disagreed with these concerns, offering the following clarifications:
- As mentioned by the other three reviewers, the dataset is of high value and the authors have provided more details in the rebuttal.
- Although the method shares conceptual similarities with prior work, the task and approach exhibit distinct innovations.
- The FDCA method was designed for composed retrieval tasks, and its performance on general benchmarks, while slightly below SOTA, remains competitive.

---

### Decision · Program_Chairs · 2025-01-22

Accept (Poster)